# The Effect of Cryopreserved Sperm on the Early Development, Survival, and Growth of Intergeneric Sterbel Hybrids (*Acipenser ruthenus* × *Huso huso*)

**DOI:** 10.3390/ijms25115784

**Published:** 2024-05-26

**Authors:** Dorota Fopp-Bayat, Anna Nitkiewicz, Piotr Gomułka

**Affiliations:** 1Department of Ichthyology and Aquaculture, University of Warmia and Mazury in Olsztyn, 10-719 Olsztyn, Poland; pgomulka@uwm.edu.pl; 2Department of Pond Fishery, National Inland Fisheries Research Institute, Oczapowskiego 10, 10-719 Olsztyn, Poland; a.nitkiewicz@infish.com.pl

**Keywords:** Acipenseridae, intergeneric hybrids, artificial reproduction, genetic identification, molecular markers

## Abstract

The aim of this study was to analyze the survival and growth of intergeneric (*Acispenser ruthenus* × *Huso huso* L.) sterbel hybrids obtained by fertilizing sterlet eggs with cryopreserved beluga semen. The rate of embryonic development did not differ between sterbel hybrids (experimental groups) and sterlets (control groups), and the hatching period was identical in all groups. The survival rate of hybrid larvae was higher in the experimental groups than in the control groups. Body weight and body length measurements revealed that sterbel hybrids grew at a faster rate than the control group sterlets. The hybrid origin of sterbels produced with the use of cryopreserved beluga semen was confirmed in a genetic analysis based on species-specific DNA fragments. To the best of the authors’ knowledge, this is the first study to analyze the growth of sterbel hybrids derived from cryopreserved semen. The research findings indicate that this type of intergeneric hybridization delivers satisfactory results and can be applied in sturgeon aquaculture.

## 1. Introduction

Fish of the family Acipenseridae are among the oldest species of freshwater fish, and this ancient lineage of vertebrates began to evolve around 200 million years ago [1]. At present, all sturgeon species living in the wild are endangered, and some species have a critically endangered status. Modern sturgeon farming and rearing techniques have been developed in aquaculture to cater to the growing demand for sturgeon meat [1,2].

Most farms supply sturgeon meat and caviar due to a high demand for these products. Some farms rear broodstocks to restore native populations of endangered sturgeon species. Most farms rear fish for human consumption, whereas only a small number of aquaculture businesses breed sturgeons under controlled conditions due to the limited availability of adequately sized broodstocks and spawners for reproductive purposes. Farms with hatchery-reared spawning stocks produce pure species of the family Acipenseridae, as well as hybrids [3,4,5].

Interspecific hybridization is a process of mating different species of the same genus. Hybrids display the traits of both parents, which enables them to better adapt to a specific environment, provided that they are capable of surviving in that environment [4,6]. Natural hybridization is ubiquitous in sturgeons, and it is often driven by fertile interspecific and intergeneric hybrids. Many sturgeon hybrids have been researched both in nature and aquaculture [7,8,9]. Intergeneric crosses between *Huso huso* beluga (genus *Huso*) and *Acipenser ruthenus* sterlets (genus *Acipenser*) produce highly interesting hybrids [10]. The hybrids derived from beluga females and sterlet males are known as besters (official international name), whereas the crosses between sterlet females and beluga males are referred to as sterbels [10,11]. Bester and sterbel hybrids play an important role in aquaculture because they inherit valuable traits from their parents, including rapid growth (inherited from belugas) and early sexual maturation (inherited from sterlets) [10,11]. In addition, these hybrids easily adapt to artificial rearing conditions and thrive on aquaculture feeds. Sturddlefish, a hybrid of the Russian sturgeon *Acipenser gueldenstaedtii* (family Acipenseridae) and the American paddlefish *Polyodon spathula* (family Polyodontidae), is probably the most exotic sturgeon hybrid [12].

Intergeneric hybridization research of fish usually include the analysis of embryonic development (during eggs incubation). This analytical approach enables identification of normal or abnormal embryo development. The embryonic development of sturgeon fish is most often monitored at four stages: (1) mitotic division I-II, (2) gastrulation, (3) neurulation, (4) prior hatching. The most developmental abnormality in sturgeon embryos results from morphogenesis during gastrulation [5,9,11]. This is usually observed in the onset of neurulation stage when the yolk plug is not covered during neural plate formation [5,9,11].

In sturgeon hybridization, fresh semen is more frequently used than cryopreserved semen. To the best of the authors’ knowledge, the applicability of cryopreserved semen for crossing sturgeon species has only been examined in the following hybrids: sterlet × Siberian sturgeon, sterlet × Russian sturgeon, sterlet × European sturgeon [13], and sterlet × beluga [11]. The semen cryopreservation procedure and fertilization success in the production of sterbel hybrids have been described in a recent study [11]. The authors analyzed the embryonic development of sterbel hybrids based on microscopic observations of incubated embryos. The embryonic development of sterbels has been thoroughly investigated, but the growth rate of juvenile hybrids has not been examined to date. Therefore, the aim of this study was to analyze the mortality and growth of larval and juvenile stages of sterbel hybrids derived from cryopreserved semen.

In this research we have chosen very interesting intergeneric hybridization of the smallest sturgeon species–sterlet (Genera–*Acipenser*) and the largest–beluga sturgeon (Genera–*Huso*); both species have 120 chromosomes [14,15]. Moreover, beluga is critically endangered and the possibility to use the cryoconserved sperm for hybrids creation is significant in innovative aquaculture.

## 2. Results

The embryos of sterbel hybrids derived from cryopreserved semen developed normally and at a similar rate to that noted in sterlet embryos in the control groups (Figure 1). No differences in embryonic growth rates were observed between the experimental groups and the control groups (Figure 1A–F). The embryos of sterbel and sterlet hybrids were analyzed in the following stages: *the* late gastrula stage (27 hpf; Figure 1A,B), *the* neurula stage (51 hpf, Figure 1C,D), cardiogenesis (72 hpf, Figure 1E,F), and directly before hatching (110 hpf, Figure 1G,H). Differences in embryonic development were observed between sterbel hybrids and sterlets only after yolk sac absorption (Figure 2). Sterbel hybrids were characterized by faster growth, visible eye pigmentation, and more advanced skin pigmentation (Figure 2). The oral cavity was also more evident in sterbel hybrids (Figure 2).

Survival analysis showed that survival in the experimental group was significantly lower (log-rank test, *p* < 0.001) when compared to control embryos (Figure 3). In contrast to embryos, *the* survival of Sterbel larvae was significantly higher (log-rank test, *p* < 0.01) (Figure 4). Lower mortality in hybrid larvae could be attributed to heterosis.

Body weight and body length were measured in 11 fish from each group during the rearing period (30 dph to 89 dph), and the results are presented in Table 1, Figure 5 and Figure 6.

The overall growth rate was highly similar in both groups (Figure 5 and Figure 6). The experimental fish were larger at the beginning of the experiment, which resulted in higher body weight gain.

The specific growth rate of experimental and control group fish during the rearing period is presented in Figure 7. Total SGR was higher in the experimental group (10.4%) than in the control group (9.1%) (Figure 7). The ability to grow quickly became apparent in Sterbel larvae after they started feeding with composed feeds.

The results of the genetic analysis of sterbel hybrids based on species-specific DNA fragments are presented in Figure 8 and Figure 9. The molecular analysis confirmed complete hybridization (100%) of sterlet and beluga hybrids derived from cryopreserved beluga semen (Figure 8 and Figure 9). In this study, we used molecular markers 153_HHp specific to beluga that was expressed in the sample extracted from beluga cryopreserved sperm (line 1; Figure 8) and hybrid offspring (lines 4–11; Figure 8) except sterlet females (lines 2–3; Figure 8). We also used molecular markers 247_Arp specific for sterlets that were expressed in sterlet females used for hybridization and sterbel offspring (lines 2–3 and 4–11, respectively; Figure 9). This marker was absent in *the* sample isolated from cryopreserved sperm from belugas (line 1; Figure 9).

Genotyping analysis of four4 microsatellite DNA markers (Spl-101, Spl-106, Spl-163 and Spl-168) was also used to confirm the hybrid origin of sterbels. This analysis showed the presence of alleles characteristic of both the female sterlets and the male beluga in the hybrids, which further confirmed the successful hybridization (Table 2). The Spl-168 locus in sterlets (parents and offspring) showed only null alleles, while in the beluga male (donor of cryopreserved semen) two alleles were identified in this locus—177 and 181 bp (base pairs). In the sterbel hybrids, either the 177 bp allele or the 181 bp allele together with *the* null allele inherited from the sterlet mother were observed. This microsatellite locus proved to be the most reliable (of the four used) for identifying sterbel hybrids. Null alleles were also identified in the Spl-104 locus, in the presence of other alleles (304 and 280 bp). The highest allelic diversity was observed at the Spl-163 locus, which was particularly visible in the sterbel hybrid genotypes.

## 3. Discussion

Interspecific and intergeneric hybridization occurs frequently in fish of the family Acipenseridae, and it is commonly used in breeding practice to obtain hybrids with desirable characteristics of both parent species [16]. The interspecific hybridization of sterlet female with beluga cryopreserved sperm resulted in obtaining of sterbel hybrids that were rearing until 89 days post hatching. The present study indicates the differences in survival of sterbel hybrids during embryonic development that was significantly lower when compared to pure sterlet control groups. But, during larvae and juvenile rearing, the survival of sterbel hybrids was significantly higher than pure sterlet individuals. The higher survival of pure sterlet species during embryogenesis may be a result of fertilization rate. Similar results were obtained in sterlet crossed with crossed with Siberian sturgeon when the lower fertilization rate was observed in the sterlet × Siberian sturgeon hybrids compared to Siberian sturgeon pure species [17].

Our study presents the significant differences in growth (length and weight) between sterbel hybrids and pure sterlet individuals. The specific growth rate (SGR) observed in sterbel hybrids (10.4%) was higher when compared to pure sterlet (9.1%). The interspecific hybridization was also conducted by Linhartová et al. [18] by mating the sterlet, beluga, Siberian sturgeon *Acipenser baerii*, and Russian sturgeon *Acipenser gueldenstaedtii* to investigate the effect of hybridization for gonadal development of experimental hybrids compared to the purebred fish [18]. Shivaramu et al. [19] compared the cumulative survival and mean body weight in the artificially produced hybrid of the sterlet and Siberian sturgeon with respect to their pure parental species. Fertilization and hatching indices were significantly higher in the Siberian sturgeon purebred compared to the sterlet × Siberian sturgeon hybrid. The highest values of body weight were identified in the sterlet × Siberian sturgeon hybrid St × S hybrid (557.54 ± 179.7 g) at 862 days post-hatch [19]. The highest positive heterosis (51.3% on 862 dph) was observed for mean body weight the sterlet × Siberian sturgeon hybrid [19].

In this study, very interesting is the research on sterbel hybrids produced with cryopreserved sperm o beluga. Application of cryopreserved sperm in aquaculture is promising and can bring many benefits, especially for fish that spawn in other seasons. The genetic material of valuable fish species is stored in gene banks to protect native populations and endangered fish species. Breeding programs involving cryopreserved semen do not require the physical presence of males, which is a considerable advantage when male and female fish attain sexual maturation in different periods. Cryopreserved semen is much easier to transport than spawners, and spawning can be carried out even when females and males are located a considerable distance apart. Cryopreserved sperm was widely used in fish breeding research at the turn of the 20th and 21st centuries [13,20,21,22,23,24,25,26,27,28]. In the following years, cryopreserved semen was used to obtain sturgeon hybrids. The first successful hybridization of sturgeons with cryopreserved sperm was performed by Urbanyi et al. [13]. In the cited study, the hatching rates of the obtained sterlet × Siberian Sturgeon, sterlet × Russian sturgeon, and sterlet × European sturgeon hybrids were determined at 50%, 17.5%, and 34%, respectively, relative to 30.6% in the sterlet control group [13].

Cryopreserved semen can be used in experiments to produce interspecific hybrids of fish that breed in different periods or even seasons of the year. The embryonic development of intergeneric hybrids was observed and described relative to sterlet embryos in the control group. In this research no developmental defects were observed in sterbel hybrids compared to sterlet control groups. Body weight and body length measurements revealed that hybrids grew at a faster rate than sterlets in the early stages of rearing. The survival rate of larvae was higher the in sterlet × beluga hybrids than in pure sterlets. In a study by Urbanyi et al. [13], survival rates were also higher in sterlet × Siberian sturgeon hybrids than in pure sterlets.

The molecular markers, especially nuclear DNA (species-specific DNA markers, SNP, microsatellite DNA fragments) allow the identification of interspecific or intergeneric sturgeon hybrids [29,30]. In the present study, the genetic analysis of sterbel hybrids confirmed successful hybridization with the use of cryopreserved semen. Genetic markers were applied to identify sterlet (parents and progeny of control groups), belugas (cryoconserved sperm), and DNA fragments characteristic of both species in sterbel hybrids. Control group sterlets carried the genetic material of the pure sterlets. Applied molecular markers for genetic identification of belugas and/or sterlets can be used in genetic identification of pure species of sterlet or beluga or their hybrids—the bester or sterbel [13,29].

Urbanyi et al. [13] used Random Amplified Polymorphic DNA (RAPD) for genetic identification of sturgeon hybrids produced with application cryopreserved semen. In this experiment all sturgeon hybrids were characterized by DNA bands of both parent species [13]. However, growth rates were not compared in hybrids and control group individuals [13]. To the best of the authors’ knowledge, there are no published data on the body length and body weight of hybrids derived from cryopreserved sperm. This is the first study to provide such information. The present findings are particularly valuable because the studied hybrids play an important role in sturgeon aquaculture.

The present paper described the applicability of cryopreserved milt for sturgeon hybridization. This type of research plays a very important role in restoring endangered populations because cryopreserved sperm of rare and valuable species/individuals can be applied to “rescue” such populations and increase genetic diversity with those valuable genomes/genotypes at both individual and population levels. Endangered populations can be reinstated through androgenesis, a reproductive process that involves cryopreserved sperm and the roe of different fish species whose genetic DNA has been inactivated, for example, through exposure to UV or gamma radiation. Grunina et al. [31] demonstrated that endangered Russian sturgeons can be conserved through androgenesis by fertilizing inactivated eggs with fresh or cryopreserved sperm of the Russian sturgeon.

In Poland, research studies evaluating the quality of fish sperm are undertaken to establish and maintain gene banks. At present, such research is conducted by the Institute of Animal Reproduction and Food Research of the Polish Academy of Sciences in Olsztyn which stores the sperm of valuable fish species, including Acipenseridae, in a gene bank. Beluga is undoubtedly the most valuable Acipenseridae species, and its sperm can be used to fertilize beluga roe and obtain genetically pure offspring. Cryopreserved sperm can also be applied to produce F1 hybrids with superior traits (rapid growth and early sexual maturation). It should be noted that beluga spawners are in short supply, and selective breeding of belugas in aquaculture is a long and costly process (belugas reach sexual maturity at the age of 18+ years). Therefore, cryopreserved semen can significantly facilitate fish breeding. In the current study, sperm for cryopreservation was harvested from only one male due to the limited availability of beluga spawners. However, sturgeon farming is booming in Poland, and in the future, sperm from a larger number of beluga males will become available for cryopreservation. At present, sperm cryopreservation is a costly procedure, but if the gene pool of Acipenseridae becomes substantially depleted in the future, stored sperm will be the only resource for optimizing the genetic diversity of sturgeons, regardless of the cost. This experiment demonstrated that cryopreservation is an effective method of preserving endangered populations of valuable fish species both in nature and in aquaculture.

## 4. Materials and Methods

This study was approved by the Local Ethics Committee for Animal Experimentation of the University of Warmia and Mazury in Olsztyn (decision No. 75/2012), and it was conducted based on the guidelines of the Animal Research Act of 21 January 2005 (Journal of Laws, 2005, item 289).

### 4.1. Gamete Collection, Artificial Reproduction and Hybridization

The experiment involved eggs collected from two sterlet females (female I and female II) aged 6+ years, fresh semen harvested from a sterlet male aged 6+ years, and cryopreserved semen collected from a beluga male aged 18+ years. Body weight was determined at 3.5 kg in sterlet female I, 4 kg in sterlet female II, 3 kg in the sterlet male, and 65 kg in the beluga male. All fish were bred under controlled conditions in the Wasosze Fish Farm near Konin, Poland.

The examined fish were subjected to environmental stimulation until the achievement of gonadal maturity stage IV [25]. Ovulation and spermiation were induced with the luteinizing hormone-releasing hormone (Sigma, Frankfurt am Main, Germany) according to the method described by Fopp-Bayat et al. [11]. Semen was collected with a syringe with an elastic cannula. Cryopreserved semen was obtained from the semen bank of the Department of Gamete and Embryo Biology of the Institute of Animal Reproduction and Food Research, Polish Academy of Sciences in Olsztyn, Poland. The procedure of cryopreserving beluga semen was described previously by Fopp-Bayat et al. [11]. Cryopreserved sperm was thawed in a water bath at 40 °C and used to fertilize sterlet eggs. Sterlet eggs were collected in vivo by abdominal massage and fertilized with cryopreserved beluga semen (experimental groups—sterbel hybrids) and fresh sterlet sperm (control groups—sterlet control).

### 4.2. Eggs Incubation

The eggs collected from two females were combined and divided into three experimental groups (E1, E2, and E3) and three control groups (C1, C2, and C3) (Figure 10). Each experimental group consisted of approximately 2000 eggs, which were fertilized with 0.5 mL of semen diluted 1:200 with hatchery water. Fertilized eggs were incubated in six incubation tanks [32] at 17 °C (Figure 10). Eggs were incubated at a constant temperature of 16 °C until hatching. The percentage of fertilized eggs was determined 4 h post-fertilization (hpf), the percentage of embryos in the neurula stage was determined 50 hpf, and the percentage of hatched eggs (emergent larvae) was determined 6 days post-fertilization (dpf). Dead embryos and embryos infected with *Saprolegnia* sp. were removed during incubation.

### 4.3. Experimental Rearing of Sterbel and Sterlet Larvae

Newly hatched larvae from the experimental (sterbel) and control (sterlet) groups were transferred to two 40 L tanks (Figure 10) in a recirculating aquaculture system (RAS) in the Center for Aquaculture and Environmental Engineering of the University of Warmia and Mazury in Olsztyn. Larvae were reared for 11 weeks. Dead fingerlings were removed, and mortality rates were registered during rearing. The applied feeding protocol was described previously by Laczynska et al. [33]. Live Artemia nauplii (origin: GSL, INVE Aquaculture, Belgium) were administered from day 7 post-hatch (dph). Larvae were fed three times per day. Artemia feeding rates were set at 50% of fish biomass in the first week, 25% of fish biomass in the second week, and 9% of fish biomass in the third week of rearing. At 38 dph, starter feed (Skretting, Perla Larvae Proactive, Stavanger, Norway) was introduced, and Artemia feeding was gradually discontinued. The live feed was completely replaced by the starter feed at 45 dph.

The survival rates of experimental and control group embryos were analyzed during incubation and rearing. The body weight and body length of experimental and control group fish were measured beginning from 30 dph. The measurements were conducted on five dates (30, 37, 44, 56, and 89 dph), separately in each of the eleven fish from each group (sterbel—experimental group; sterlet—control group). Before each measurement, fish were sedated with a solution of clove oil (0.4 mg/L). After the measurements, fish were transferred to a tank with fresh water for recovery. Body weight was determined with a Sartorius AG analytical balance (Sartorius, Gottingen, Germany) laboratory scale to the nearest 0.01 g, and body length (Longitudo Totalis) was measured with a ruler to the nearest 1 mm. During the experiment, the specific growth rate (SGR) was determined on four dates with the use of the following formula: SGR = [(lnW2 − lnW1)/t] × 100%, where W1 is the mean initial body weight, W2 is the mean final body weight, and t is the time between measurements W1 and W2.

### 4.4. Genetic Analysis

Two nuclear DNA fragments characteristic of sterlet (247_Arp) and beluga (153_HHp) genotypes were collected from ten randomly selected sterbels and sterlets for genetic analysis [29]. Nuclear DNA fragments from two sterlet females and one sterlet male, as well as cryopreserved semen were also used in the genetic analysis. DNA was isolated from fin fragments sampled from hatched larvae after yolk sac absorption. Genomic DNA was isolated from cryopreserved semen according to the method described by Fopp-Bayat and Ciereszko [34]. DNA was isolated from fins with the use of the Sherlock DNSA purification kit (A&A Biotechnology, Gdańsk, Poland) according to the manufacturer’s instructions. DNA fragments 247_Arp and 153_HHp were amplified as described by Havelka et al. [29]. The amplified DNA fragments were separated by electrophoresis on 2.0% agarose gel with the addition of ethidium bromide and analyzed under UV light with the use of the G:Box gel doc system (Syngene, Cambridge, UK) and GeneSnap 7.09 software (Syngene, Cambridge, UK). The DNA of hybrid progeny in the experimental groups was compared with the parental genotypes (sterlet and beluga).

Twenty randomly selected sterbels hybrid and twenty sterlets from the control group were analyzed using four microsatellite DNA markers: Spl-101, Spl-106, Spl-163 and Spl-168 [35] based on the procedure described by Fopp-Bayat et al. [11]. The analysis of amplified microsatellite DNA loci was conducted using an Applied Biosystem 3130 Genetic Analyzer with the application of 5′-labeled with the different fluorescent reporter dyes forward primers. Genotyping of the microsatellite DNA alleles was conducted using an Applied Biosystems 3130 Genetic Analyzer sequencer against the GeneScan 600 [LIZ] size standard (Applied Biosystems, CA, USA). The size of microsatellite DNA fragments was determined using the Genetic Analyser software GeneMapper v4.1 (Applied Biosystems, CA, USA) according to the manufacturer’s recommendations.

### 4.5. Statistical Analysis

Data are presented as mean ± SD. Due to the lack of normality of distribution (Shapiro-Wilk test, *p* < 0.05) and homogeneity of variances (F test, *p* < 0.05), the non-parametric Mann-Whitney test was used for comparisons between groups at the same time points in case of length and weight. A log-rank test was used for survival analysis. Differences in survival inside groups were tested by a Chi2 test. Statistica 13 software (TIBCO Software Inc., Palo Alto, CA, USA) was used for statistical analysis of survival and growth in experimental and control groups. The Student’s *t*-Test (*p* < 0.0001) was used for analysis of the length of sterbel hybrids and sterlet control specimens (aged 10 dph) with application GraphPad Prism10 software.

## 5. Conclusions

This is the first study to analyze the growth of sterbel hybrids derived from cryopreserved semen. Sterbel hybrids obtained by fertilizing sterlet roe with cryopreserved beluga sperm are characterized by faster growth rates and higher survival rates than pure sterlets. The described methodology and the results of experimental studies confirm that cryopreserved sperm can be successfully used to produce valuable sterbel hybrids in sturgeon farming.

## Figures and Tables

**Figure 1 ijms-25-05784-f001:**
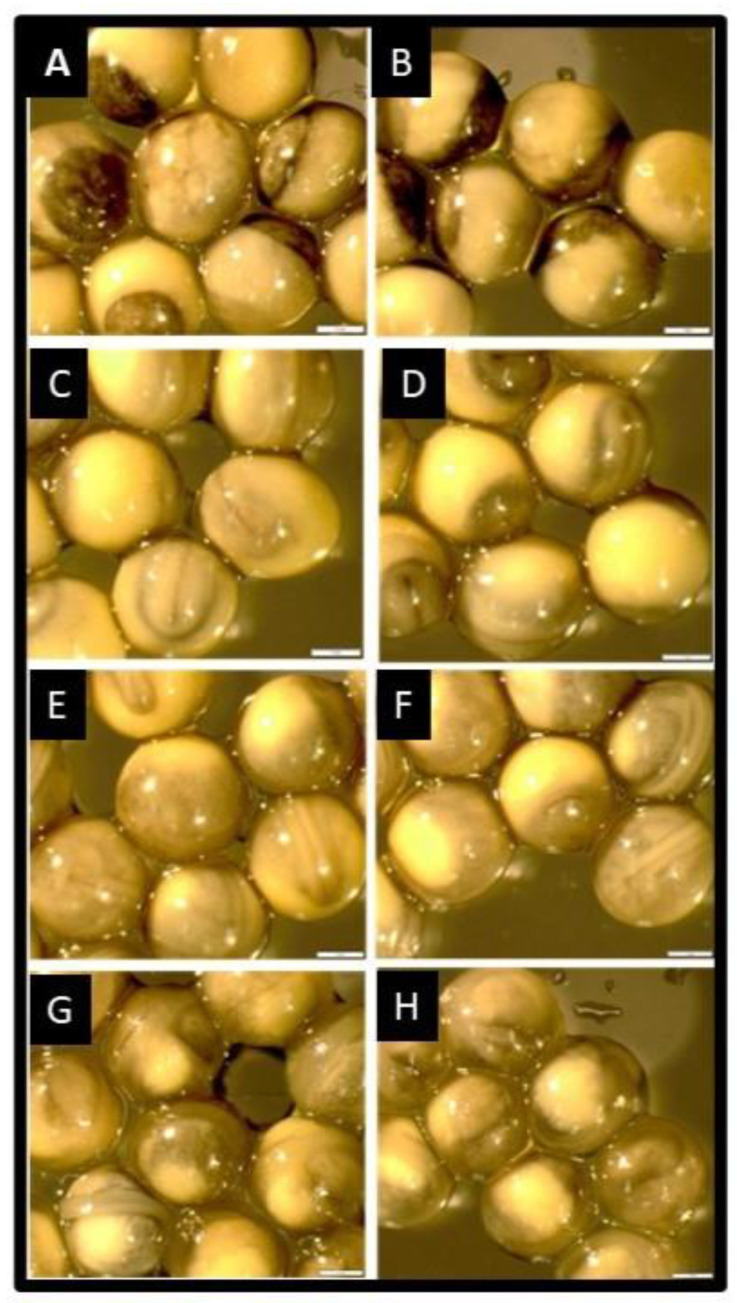
Embryonic development of a sterbel hybrid (*A. ruthenus* × *H. huso*)*,* left column, and a sterlet (*A. ruthenus*)*,* right column; (**A**,**B**)—late gastrula stage (27 hpf); (**C**,**D**)—neurula stage (51 hpf); (**E**,**F**)—cardiogenesis (72 hpf); (**G**,**H**)—directly before hatching (110 hpf). Scale bar—1 mm.

**Figure 2 ijms-25-05784-f002:**
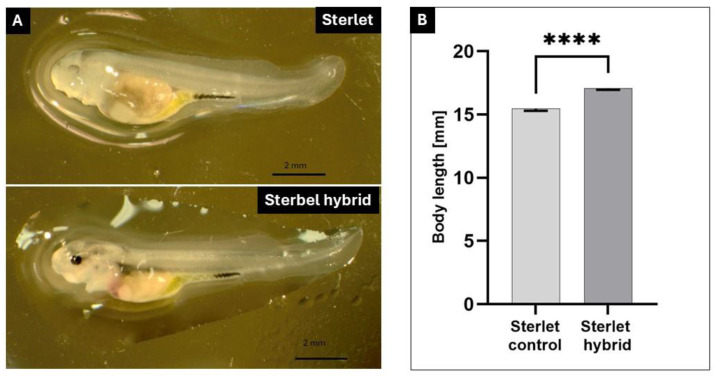
Sterbel hybrid (*A. ruthenus* x *H. huso*) and sterlet (*A. ruthenus*) larvae after yolk sac absorption. (**A**)—sterlet; (**B**)—sterlet × beluga. Scale bar—2 mm; Part A. Length of Sterbel hybrid (*A. ruthenus* × *H. huso*) and sterlet (*A. ruthenus*) larvae in 10 day post hatching (dph); **** *p* < 0.0001; Part B.

**Figure 3 ijms-25-05784-f003:**
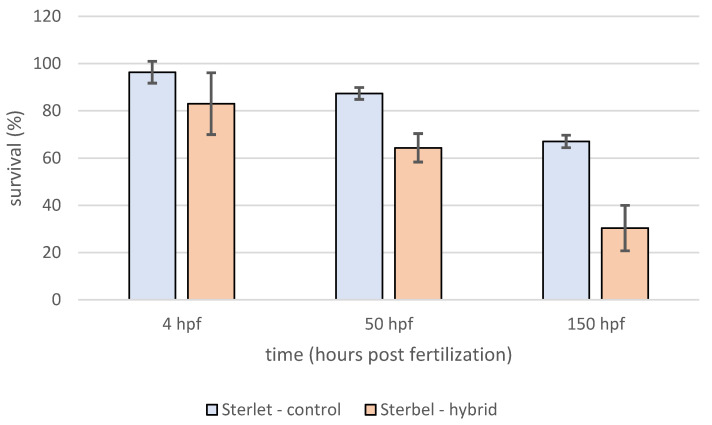
Survival of Sterlet—control and Sterbel—hybrids embryos during the experiment. Bars represent standard deviation.

**Figure 4 ijms-25-05784-f004:**
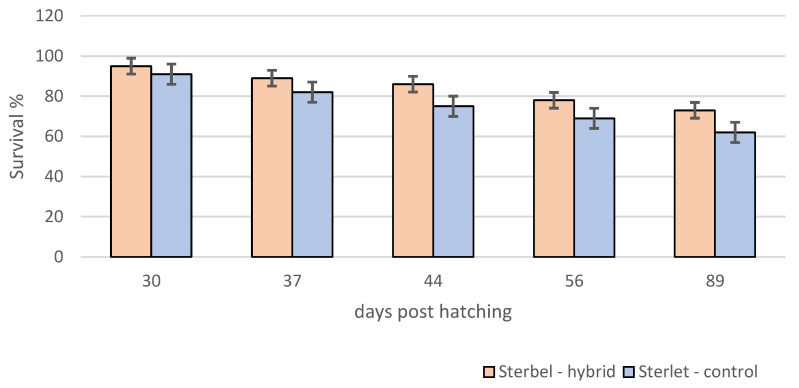
Survival of sterbel—hybrids and sterlet—control larvae during experimental rearing. Bars represent standard errors.

**Figure 5 ijms-25-05784-f005:**
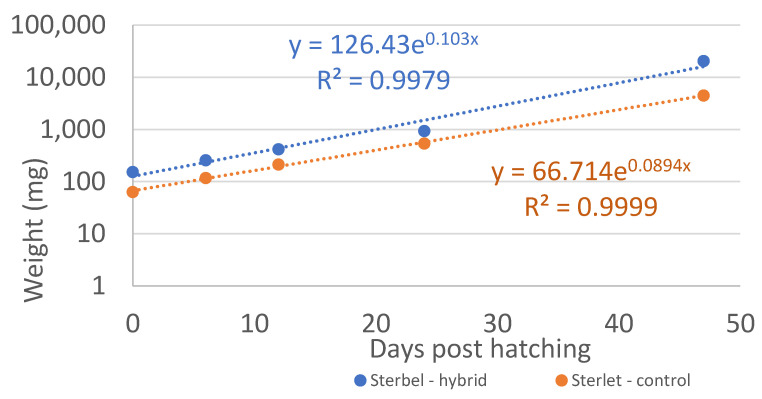
Body weight of Sterbel hybrids during experimental rearing relative to the Sterlet—control group.

**Figure 6 ijms-25-05784-f006:**
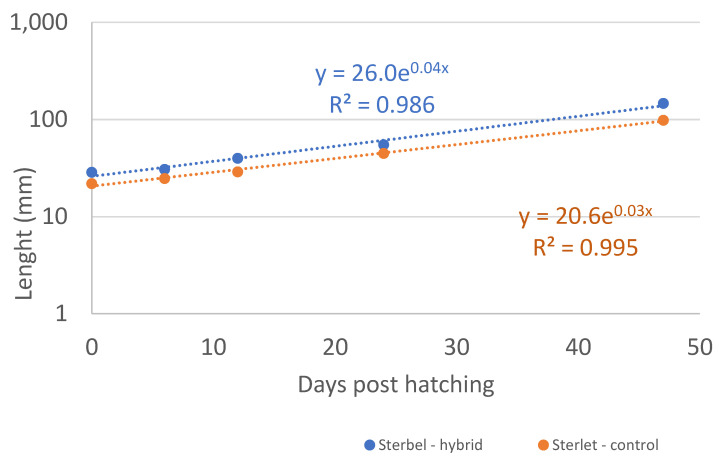
Body length of Sterbel-hybrids during experimental rearing relative to the Sterlet—control group; sterbel hybrid—blue; control group sterlet—orange.

**Figure 7 ijms-25-05784-f007:**
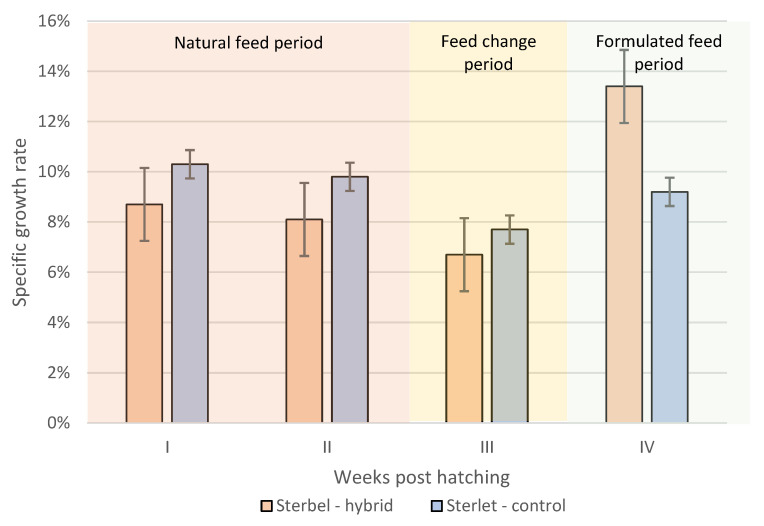
Specific growth rate (SGR) in the experimental group (Sterbel-hybrid) and the control group (Sterlet-control). I—30–37 dph, II—37–44 dph, III—44–56 dph, IV—56–89 dph. Bars represent standard errors.

**Figure 8 ijms-25-05784-f008:**
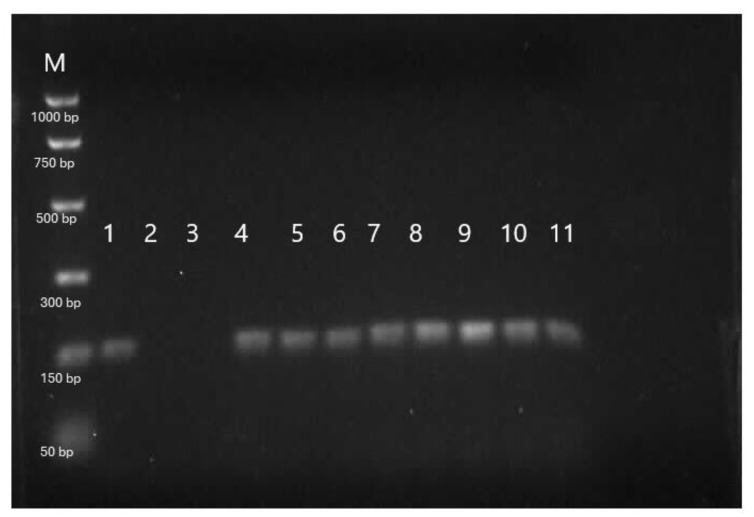
Genetic analysis of a sterbel hybrid with the use of species-specific DNA fragments for belugas (153_HHp). M—molecular pattern, 1—beluga male, 2–3—sterlet females, 4–11—hybrid offspring (sterbel). M—PCR Markers (Promega).

**Figure 9 ijms-25-05784-f009:**
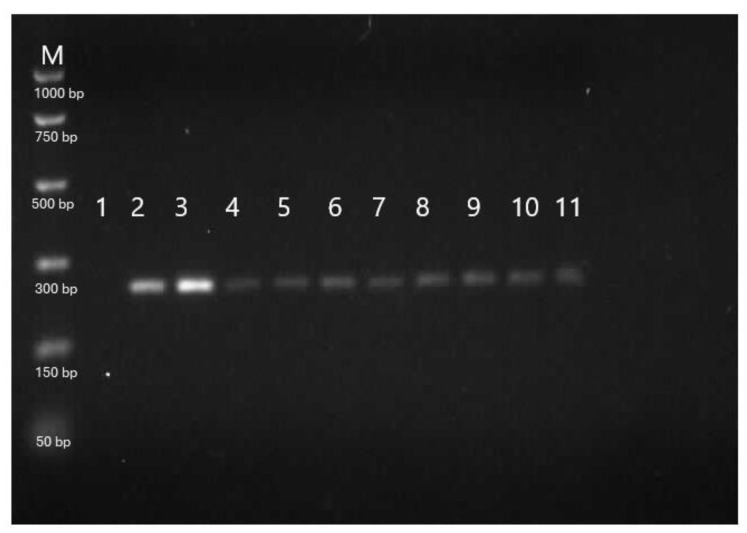
Genetic analysis of a sterbel hybrid with the use of species-specific DNA fragments for sterlets (247_Arp). M—molecular pattern, 1—beluga male, 2–3—sterlet females, 4–11—hybrid offspring (sterbel). M—PCR Markers (Promega).

**Figure 10 ijms-25-05784-f010:**
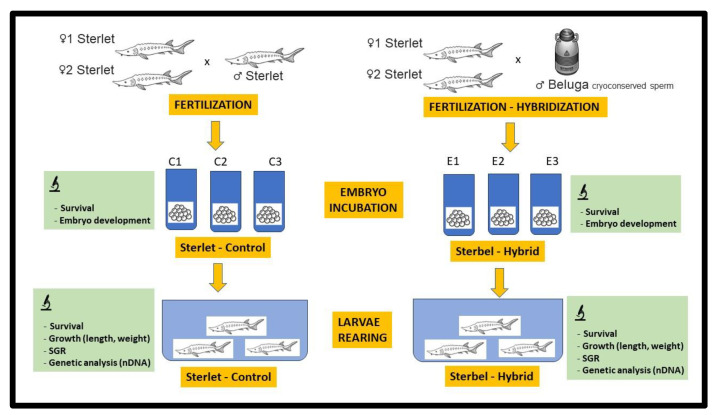
The graphical experimental design.

**Table 1 ijms-25-05784-t001:** Body weight and body length measured during experimental rearing between 30 dph and 89 dph.

Fish Age in dph	Length (mm)		Weight (mg)	
	EXP	CON	EXP	CON
30	28.6 ± 1.0 ^a^	21.9 ± 1.6 ^b^	152 ± 18 ^a^	63 ± 16 ^b^
37	30.6 ± 1.9 ^a^	24.7 ± 0.9 ^b^	257 ± 56 ^a^	117 ± 15 ^b^
44	39.8 ± 2.4 ^a^	28.9 ± 2.1 ^b^	416 ± 35 ^a^	212 ± 61 ^b^
56	55.1 ± 2.8 ^a^	44.8 ± 6.3 ^b^	932 ± 88 ^a^	535 ± 195 ^b^
89	147.0 ± 7.5 ^a^	98.3 ± 10.2 ^b^	20,414 ± 1860 ^a^	4491 ± 1535 ^b^

In Table 1: values in the rows denote mean body length and mean body weight. Values marked with different superscript letters differ significantly at *p* < 0.001. a, b indicate statistically significant differences between the analyzed data.

**Table 2 ijms-25-05784-t002:** Alleles segregation in studied microsatellite DNA loci of sterbel hybrid offspring and sterlet (female) and beluga (male) parents.

Parents/Offspring Genotypes	Microsatellite DNA Locus			
	*Spl-101*	*Spl-106*	*Spl-163*	*Spl-168*
SF1; Sterlet–Female 1	304/304	240/314	188/206	null/null
SF2; Sterlet–Female 2	304/304	188/206	188/206	null/null
BM; Beluga–Male (cryopreserved sperm)	276/276	180/196	180/196	177/181
SM; Sterlet–Male control	280/280	240/272	188/206	null/null
SF1 × SM	304/null 280/null 280/304	240/240 240/272 240/314 314/272	188/188 188/206 206/206	null/null
SF2 × SM	304/null 280/null 280/304	188/240 206/240 188/272 206/272	188/188 188/206 206/206	null/null
SF1 × BM	276/304 304/null	180/240 180/314 196/240 196/314	180/206 196/206 180/188 188/196	177/null 181/null
SF2 × BM	276/304 304/null	180/188 180/206 188/196 188/206	180/206 196/206 180/188 188/196	177/null 181/null

## Data Availability

The raw data supporting the conclusions of this article will be made available by the authors upon request.

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
