# Peer review of "The Effect of Cryopreserved Sperm on the Early Development, Survival, and Growth of Intergeneric Sterbel Hybrids (Acipenser ruthenus × Huso huso)"

_ijms, 2024, doi:10.3390/ijms25115784_

Round 1

Reviewer 1 Report

Comments and Suggestions for Authors

Please, find my comments in the attachment.

Author Response

Dear Reviewer,

We would like to thank for review of our paper. All suggestions are very valuable for us and the paper was corrected according the suggestions included in reviews.

The response to reviews was provided below.

Review 1

  1. This research aims to study the production of sturgeon hybrids by analysing the effects of the use of cryopreserved sperm on growth and mortality as well as larval deveolpment, in comprison to fresh sperm.

The introduction is quite clear and well organised but also very short and simple, it lacks relevant references, as the authors used only 5 studies as reference. This must be further developed. They should also include information about embrionic development to put in context what they measure later on, among other aspects.

Ad. 1. We completed the Introduction section with additional references according to Reviewer’s suggestions. We also included the paragraph (attachet below) regarding embryonic development, as it was suggested.
“Intergeneric hybridization research of fish usually include the analysis of embryonic development (during eggs incubation). This analytical approach enables identification of normal or abnormal embryo development. The embryonic development of sturgeon fish is most often monitored at four stages: 1) mitotic division I-II, 2) gastrulation, 3) neurulation, 4) prior hatching. The most developmental abnormality in sturgeon embryos results from morphogenesis during gastrulation [5,9,11]. This is usually observed in the onset of neu-rulation stage when the yolk plug is not covered during neural plate formation [5,9,11]”.

  1. Authors mention that they aim to study differences between sterbels generated with fresh or cryopreserved sperm, nevertheless they also compare the embryonic growth and development with sterlets, which is a bit confusing. They do this comparison again, in different parts of the manuscript. They should include this as one of the aims, explaining the interest of doing it, or either, remove this particular data comparing the 2 types of hybrids which appsrently, was not the aim of the study. If they decide to still keep the data, it has to be well organised. At the moment there is a mixture of figures that compare both types of hybrids and the ones that compare fresh and cryopreserved sperm.

Ad. 2. The aim of the present study was the comparison of sterbel hybrids (produced with cryopreserved sperm-only) with pure sterlet. We compared the embryonic development of sterbel (hybrids) to sterlet (pure species) to check if sterbel embryo development is proper (without defects). Unfortunately, we do not produce the sterbel hybrids with fresh sperm, because we did not have a fresh fresh sperm of beluga. We produced only one kind of hybrids that was presented in the additional Figure 10. The graphical experimental design (the scheme of experiment).

  1. Besides, the figures (1 and 2) lack a scale or similar that shows the size of the embryo stages. Authors mention differences in survival showed in figure 3, including some statistical analysis but nevertheless, they don't show error bars in the figure, nor in figure 4 either.

Ad. 3. The scale was included in Figures 1 and 2. We also included the statistical analysis that correspond with Figure 1 based on the length of sterbel and sterlet larvae in the age of 10 days. We also completed the Figure 3 with standard deviation and Figure 4 with standard errors as it was suggested.

  1. Figured 8 and 9 are also very confusing. The size of the ladder is not included and the author mention sizes at some parts of the text, which, without a ladder, is impossible to prove. In general, all the ?igures must be improved and include the relevant information to be able to analyse the data properly.

Ad. 4. We included the size of the ladder in the Figure 8 and 9 during Reviewer’s suggestions.

  1. The discussion is quite messy, mixing some parts that could be included in the introduction as well, and moreover, the text doesn't follow a logical organization or structure, it must br rewritten.

Ad. 5. We have rewritten and corrected the Discussion as it was suggested.

  1. Similarly, the material and methods is also very messy and it is not clear how the experiments were done. There is a mix including semen from.different individuals to use as fresh or cryopreserved, the 2 types of hybrids, etc. All very confusing...Furthermore, material and methods lacks sections and headings to organise the information. They have including only a section forstatistic analysis. I suggest the authors to read scienti?ic texts and look a bit more into the format to improve the quality of their manuscript, which, at the moment, is very poor.

Ad. 6. The Material and Methods section was rewritten and corrected, and the sections and headings were added. Additionally, we included the scheme of the experiment for a better understanding of the experiment design (Fig. 10).

Thank you for all suggestions.

Yours faithfully,

Dorota Fopp-Bayat

Reviewer 2 Report

Comments and Suggestions for Authors

This study examines the effect of cryopreserved sperm on the early development, survival, and growth of intergeneric sterbel hybrids. The authors performed a comprehensive range of analyses. Their results suggest that cryopreserved sperm can be applied in sturgeon aquaculture. This is an interesting result, and the experimental methods appear to be comprehensive and done well (although it should be divided into several subsections). However, I found that the manuscript is not well written. The discussion in particular is long and difficult to read. The authors need to emphasize their main findings, or what is novel about them at all. I think the manuscript would need a thorough re-write before it could be considered for publication.

Major comments:

1. The introduction is too long and redundant. The introduction should focus on your research topic, and the introduction lacks literature citations.

2. After reading the introduction, I still don't know why you chose Acipenser ruthenus and Huso huso for interspecific hybridization.

3. As mentioned above, this discussion is kind of like a review, the authors need to emphasize their main findings, or what is novel about them at all in the discussion.

4. The experimental methods is comprehensive, but the authors need to divided it into several subsections.

Minor comments:

1. Fig. 2, it is necessary to make statistics on these characteristics, so as to reflect the differences in embryonic development more easily.

2.  Fig. 3. 4 and 7, these figures don't look like they were made with professional software, error lines need to be shown, and replicate numbers need to be supplemented.

3. Fig. 5 and 6, the two broken lines are too close to each other.

4. Fig. 8 and 9. the molecular size on the marker needs to be noted.

5. The reference format is not uniform and is rather messy.

6. Use a decimal point instead of a comma in a number.

Comments on the Quality of English Language

Introduction and Discussion need to be improved. 

Author Response

Dear Reviewer,

We would like to thank for review of our paper. All suggestions are very valuable for us and the paper was corrected according the suggestions included in reviews.

The response to reviews was provided below.

Review 2.

General comments

This study examines the effect of cryopreserved sperm on the early development, survival, and growth of intergeneric sterbel hybrids. The authors performed a comprehensive range of analyses. Their results suggest that cryopreserved sperm can be applied in sturgeon aquaculture. This is an interesting result, and the experimental methods appear to be comprehensive and done well (although it should be divided into several subsections). However, I found that the manuscript is not well written. The discussion in particular is long and difficult to read. The authors need to emphasize their main findings, or what is novel about them at all. I think the manuscript would need a thorough re-write before it could be considered for publication.

The Discussion section was shortened as it was suggested. Some parts of text were deleted. We emphasized the main findings: 1) hybridization – production of sterbel with information about survival and growth, 2) application of cryopreserved sperm in reproduction and  hybridization of sturgeons, 3) application of genetic markers for hybrid identification 4) general application of  cryopreserved sperm in conservation of sturgeons, 5) general application of cryopreserved sperm of beluga based on Polish gene bank.

We rewritten the manuscript as it was suggested.

Major comments:

  1. The introduction is too long and redundant. The introduction should focus on your research topic, and the introduction lacks literature citations.

Ad. 1. The Introduction section was corrected and literature cidations were added as it was suggested.

  1. After reading the introduction, I still don't know why you chose Acipenser ruthenus and Huso huso for interspecific hybridization.

Ad. 2. In this research we have chosen very interesting intergeneric hybridization of the smallest sturgeon species – sterlet (Genera – Acipenser) and the largest – beluga sturgeon (Genera Beluga); both species have 120 chromosomes. Beluga sturgeon is the largest aquaculture fish in Europe (sexually mature beluga is 120-200 cm) with delayed maturation – spawners age is of 11 - 21 years old (Birstein 1993, Billard and Lecointre 2001). In contrast to the beluga, sterlet reach sexual maturity at 3-7 years old with body length of approx. 90 - 100 cm (Birstein 1993, Billard and Lecointre 2001). Moreover, beluga is critically endangered and the possibility to use the cryoconserved sperm hybrids for creation is significant in innovative aquaculture.

This explanation was also included in the text of the Introduction section.

  1. As mentioned above, this discussion is kind of like a review, the authors need to emphasize their main findings, or what is novel about them at all in the discussion.

Ad. 3. The Discussion section was shortened, corrected and the main finding were discussed. In this section we focused on three main aspects: interspecies hybridization, the use of cryopreserved semen for hybrid production, and genetic analyzes for hybrid identification. At the end importance of application of cryopreserved sperm banks in the reproduction and hybridization of sturgeon fish was described.

  1. The experimental methods is comprehensive, but the authors need to divided it into several subsections.

Ad. 4. The Material and Methods was divided in sections as it was suggested.

Minor comments

  1. 2, it is necessary to make statistics on these characteristics, so as to reflect the differences in embryonic development more easily.

Ad. 1. Fig. 2 – the statistics was included in this part of paper and additional part of figure was presented to confirm differences in sterbel and sterlet larvae (10-day-old).

  1. 3. 4 and 7, these figures don't look like they were made with professional software, error lines need to be shown, and replicate numbers need to be supplemented.

Ad. 2. Fig. 3. 4 and 7 were corrected during Reviewer’s suggestions. In Figures 3,4 and 7 standard deviation or standard error lines were  included.

  1. 5 and 6, the two broken lines are too close to each other.

Ad. 3. Regarding the Figures 5 and 6, the exponential trend line showed the best fit to the data presented in these graphs. Due to the use of a logarithmic scale on the Y axis (which "straightens" the exponential curve), the dashed lines cannot be moved apart. In this case, changing the minimum value on the Y axis from 1 to 10 will not help much.

  1. 8 and 9. the molecular size on the marker needs to be noted.

Ad. 4. The molecular sizes of the marker were included in the Figures 8 and 9 during Reviewer’s suggestions.

  1. The reference format is not uniform and is rather messy.

Ad. 5. The References were checked and corrected as it was suggested.

  1. Use a decimal point instead of a comma in a number.

Ad.6. We used the decimal point instead of a comma in a number as it was suggested.

Yours faithfully,

Dorota Fopp-Bayat